# Valuing Outpatients’ Perspective on Primary Health Care Services in Greece: A Cross-Sectional Survey on Satisfaction and Personal-Centered Care

**DOI:** 10.3390/healthcare12141427

**Published:** 2024-07-17

**Authors:** Dimitris Charalambos Karaferis, Dimitris A. Niakas, Dimitra Balaska, Angeliki Flokou

**Affiliations:** 1Department of Health Economics, Medical School, National and Kapodistrian University of Athens, 11527 Athens, Greece; dimitris.niakas@gmail.com; 2Department of Business Administration, University of West Attica, 12241 Athens, Greece; 3School of Social Sciences, Hellenic Open University, 26335 Patra, Greece

**Keywords:** primary care, patient-centered care, communication skills of staff, health care quality improvement, continuity of care, access, Greece

## Abstract

Introduction: The aims of the study were to identify and analyze the determinants associated with outpatient satisfaction in Greek primary care. This is because there is a general consensus that primary care is the linchpin of effective person-centered care delivery. Methods: A cross-sectional survey was conducted with 1012 patients’ exit interviews; sociodemographic variables were included in the questionnaire to obtain data on the satisfaction of primary care users with 20 public primary healthcare centers in Athens between June 2019 and April 2021. Statistical analysis was applied to 55 items and eight dimensions of patient satisfaction, namely, arrival and admission, waiting before the appointment, cleanliness of toilets, medical examination and behavior of physician, behavior of nursing staff, laboratories, departure, and contribution of the PHCs. Descriptive analyses and multiple linear regression were used to analyze the factors influencing patient satisfaction through coefficients (β) with 95% confidence intervals and associated tests of statistical significance. Results: Τwo-thirds (74.21%) of this survey’s participants ranged from 45 to 74 years of age. More than half of the participants were women (62.15%). The most common reasons for visits were pathological (26.48%), followed by cardiological conditions (9.78%), orthopedics (9.49%), gynecologic conditions (8.70%), and ophthalmologic problems (7.31%). In the center of satisfaction with primary care was the medical care and the behavior of the physician (β = 0.427; *p* < 0.01), followed by the time during appointment (β = 0.390; *p* < 0.01). Dimensions like “accessibility and availability, 2.19/5”; “waiting times, 2.89/5”; “infrastructure of facilities (2.04/5) and cleanliness of them, (2/5)”; “laboratories, 2.99/5” and “bureaucracy in the departure, 2.29/5” were crucial for the trust and satisfaction of patients. Overall satisfaction was rated at a moderate level (2.62 ± 0.18) while person-centered care was rated as weak (2.49 ± 0.28). Conclusions: Greece is recommended to increase the sensitivity of the use of the primary health care system by patients as a first contact, continuous, comprehensive, and effective patient- and family-focused care.

## 1. Introduction

Primary health care (PHC) has long been the frontline of entry for individuals into the response system, where the majority of the population’s curative and preventive health care needs can be satisfied without distinguishing social status or other factors [1,2]. A team of physicians, nurses, midwives, and other health care professionals collaborate with patients and their families to provide comprehensive and holistic solutions, ensuring that everyone has access to quality health care services and putting the specific needs of the community first, with the purpose of improving their general health [3]. Primary care services include a wide range of medical diagnostic and laboratory services, dental care, services for major infectious diseases, for chronic diseases, urgent care, and preventive services such as screening, with an emphasis on health education, counseling, and community involvement. Therefore, the significance of primary care practice is unique and an essential component of a high-quality health system, which strives to provide access to a wide range of health issues, as providing up-to-date, continuous, and patient-centered care where treatment is focused on prevention and commenced before severe problems develop [4,5,6,7].

Across developed and developing countries, quality in health care continues to gain momentum as a result of the establishment of quality improvement initiatives, quality assurance, and patient agendas. Over the past 30 years, patients’ satisfaction has become an attribute of quality and a renowned standard to evaluate the effectiveness of the health care system, since it provides as a useful insight into public opinion on health care system performance and responsiveness. Systematic reviews of published literature confirm that enhanced patient satisfaction not only results in improved patient experiences but also aligns with better treatment outcomes and lower health inequalities [8,9,10,11,12,13]. In this context, access and availability of providers as well as the communication skills of employees were identified as the main ingredients to maintain and improve the performance and outcome of health care services. Numerous studies have consistently highlighted the critical role played by primary care centers in influencing health outcomes, particularly in making fewer preventable emergency department visits and hospital admissions, enhancing the performance of the health care system, and achieving better health outcomes, especially for those who are economically disadvantaged. In countries with robust primary care, the health care system performs better, which makes it a potential solution addressing various health care system challenges [14,15,16,17,18,19,20].

Greece follows an organizational model for structuring health care services that allows direct access to the health system, so patients are free to visit either a general practitioner first or a specialist directly. Furthermore, the orientation of the NHS (ESY) was hospital-centered and completely medical-centered, aiming at the treatment of the disease and not at prevention. Preliminary results showed problems associated with access to care, management of chronic diseases, non-24-h operation of the primary health care units, use of electronic health records, bureaucracy, lack of information, and problems with basic procedures as there was no gatekeeping system. Stimulated by worsening economic effectiveness of the health care system, the number of home visits has been steadily decreasing while varying significantly between health centers, resulting in the rising prevalence of chronic diseases and leading to a high demand for secondary care, as well as to a high use of medical drugs. For this reason, significant changes in primary care centers were initiated in the early 2010s, as many aspects needed to be improved [21,22,23,24]. In 2014, as the National Organization for the Provision of Health Services (EOPYY) had the responsibility of providing primary care and at the same time was the purchaser of these services, it was deemed appropriate to separate these responsibilities and to integrate the services of primary health care into the NHS and to set up the national PHC network (PEDY) under the responsibility of the Health Authorities (DYPE) by geographical area. A new PHC model needed to be developed based primarily on family medicine—a medical specialty with a wide range of competencies, similar to the solutions found in countries with strong PHC. Thus, the new scheme formed included two levels, the first level included the PHC network and the second level included new Local Health Units (TOMYs) [25,26,27,28].

A patient’s evaluation of their medical care experience at primary care practices is impacted by their interactions with health care providers and the available resources during their visit. Therefore, measuring the quality of care provides information about the provider’s success in meeting patients’ values and expectations, which is crucial for health institutions to measure service quality in order to determine whether their services meet the needs of citizens, their functionality, and stability [29,30]. The focus of the current study was to obtain in-depth information from patients about their perceptions of care and evaluate eight key factors that are commonly associated with the patient experience of PHC network (PEDYs), namely, arrival and admission, waiting before the appointment, cleanliness of toilets, medical examination and behavior of physician, behavior of nursing staff, laboratories, departure, and the contribution of the PHC center, to provide information for the following: (1) how do Greek citizens perceive the quality of primary health care provided, (2) describe patient experiences with primary care, and (3) examine the association between primary care practices and patient satisfaction. Importantly, this study represents the first attempt to fill the gap in our knowledge after the restructuring of the national PHC network (PEDY) by examining factors associated to patient satisfaction in relation to public primary health care facilities in Greece and other health systems in the European Union.

## 2. Materials and Methods

### 2.1. Instrument

The present study is based on the questionnaire prepared in Greek by Aletras, Zacharaki, and Niakas (2007) and was conducted on a sample of 100 adults attending the regular outpatient clinic of the ophthalmology clinic of Larissa University Hospital in the first half of 2005. The response rate was 70.40%. Individual item scores were factor analyzed in order to form appropriate summated scales on outpatients’ satisfaction, the reliability and validity of which were subsequently assessed statistically. Principal components analysis revealed four factors, namely, “physician care”, “nursing care”, “appointment time”, and “environment” [31]. The satisfaction with outpatient service (SWOPS) questionnaire is a multi-dimensional outpatient instrument, can be used in any clinical setting from outpatient to inpatient and ward settings, and is a useful tool in primary care to monitor the quality of care. Two additional parts were developed after an extensive review of literature on the concept of patients’ satisfaction with the use of health services.

In Part A, we tried to retrieve information from patients’ experiences in the PEDY network of the 1st Health Region of Attica. Forty-six questions were included, most of them being closed Likert-type, with a 6-point intensity scale from strongly disagree to strongly agree and taking numerical values: 1 = strongly agree, 2 = agree, 3 = neutral, 4 = disagree, 5 = strongly disagree, 6 = don’t know/no answer. Eight dimensions were analyzed, more specifically, 11 questions (1–11) related to arrival and admission, 6 questions (12–17) related to waiting before the medical appointment, 1 question (18) related to the cleanliness of toilets, 11 questions (19–29) related to the medical examination and behavior of physician, 2 questions (30–31) related to the behavior of nursing staff, 6 questions (32–37) related to the laboratories, 4 questions (38–41) about the departure, and 5 questions (42–46) about the contribution of the PEDY network on a patient-centered basis and an overall evaluation of system responsiveness. Questions had statements positively or negatively worded in order to rule out positive bias. In the said questionnaire, sixteen (16) questions—items 2, 5, 7, 13, 15, 16, 20, 22, 24, 26, 28, 31, 33, 35, 37, and 39—were reversed before the statistical analysis, since they were worded opposite to the others. Part Β of the survey included socio-demographic data, like patients’ gender, age, marital and educational status, professional category, nationality, and economic situation. A pilot study was conducted, using a convenience sample of 25 outpatients who responded to the questionnaire in order to check the reliability of the measuring tool used.

### 2.2. Ethical Permission

The Ethical Committee of the National and Kapodistrian University of Athens approved the study protocol (1819023327-25/2/2019). Approval was also obtained from the 1st Regional Health Authority of Attica (approval number: 31714-7/6/2019). The confidentiality of the data used has been maintained according to the Hellenic Authority for Data Protection. Finally, the research process is consistent with the Helsinki Declaration of 2013 [32].

### 2.3. Study Design and Participants

The Attica region is the largest region of Greece, with a total population of around 3.75 million, approximately 35% of total Greece population. According to the Greek Ministry of Health in the year 2021, the 57 PEDYs of Attica region were supported by a total of 3690 employees (1503 medical staff, 1176 nursing staff, and 1011 other staff) and 724 medical machines for providing health services to 2.24 million appointments (1.31 million regular cases or 58.42%, 0.68 million urgent cases or 30.50%, 0.22 million prescription cases or 9.80%, and 0.028 million other cases) [33].

The survey was conducted between June 2019 and April 2021, in 20 PEDY out of a total of 57 in the 1st Regional Health Authority of Attica, provided that the PEDYs will not be presented by name. The research design was a cross-sectional analysis and the sampling technique used was convenience sampling. This is a non-probability sampling method and samples are drawn from a group of patients that are easy to contact or reach (i.e., those who were more willing to participate in the survey). The questionnaires were distributed and obtained from the patients by the researcher himself on discharge from the primary health care centers after the way and time of completing the questionnaires and the importance of their participation in the study were explicitly explained to them [34]. Furthermore, the patients and their attendants were informed of their participation in collecting information to assess the quality of care, and the survey was anonymous and voluntary, the patients had the option not to participate or to withdraw from the study at any time. Individuals had to be 18 years of age and older, know the Greek language, and be able to communicate and speak with other people. No foreigners and refugees were included.

### 2.4. Statistical Analysis

Descriptive statistics were used to report respondents’ level of patients’ satisfaction, including mean scores (ms), standard deviations (sd), and median (interquartile range). Moreover, the categorical variables were presented as absolute (N) and relative (%) frequencies. The Kolmogorov–Smirnov test was used for normality assessment. To compare the overall satisfaction score as a quantitative variable between three or more different groups, we used the ANOVA method and a non-parametric Kruskal–Wallis test. While comparing the overall satisfaction score between two groups, we used the *t*-test and non-parametric Mann–Whitney test. The individuals’ responses to the ‘don’t know/don’t want to answer’ options in the questionnaire were not included for the evaluation of acceptability [31,35,36]. Reliability analysis included Cronbach’s alpha for internal consistency. Subsequently, multiple linear regression models were used to predict the factors associated with outpatients’ satisfaction with health care service quality. Measures expressed as coefficients beta (β) with a 95% confidence interval were used to describe the association among the variables. Statistical significance was set at a two-sided *p*-value < 0.05. All statistical analyses were performed using SPSS 26.0 (IBM Corp., Armonk, NY, USA).

## 3. Results

### 3.1. Normality and Reliability Analysis

Kolmogorov–Smirnov and Shapiro–Wilk tests were applied to assess the normality of their distributions. These tests showed a statistically significant deviation from normality. Internal consistency, and reliability of the satisfaction questionnaire were analyzed, and Cronbach’s alpha coefficient was found to be >0.76, which indicates excellent reliability of the questionnaire [37].

### 3.2. Analysis of Sociodemographic and Other Characteristics of Study Participants

A total of 1012 outpatients took part in this research and were included in our analyses. The response rate was 84.33% (1012/1200). The majority of patients were women (629 or 62.15%), while the majority of the sample (751 or 74.21%) was between 45 and 74 years old. Almost two-thirds of the patients (730 or 72.13%) were married, half of the sample had a high school diploma (534 or 52.77%), and only 1.07% had a master’s or doctoral degree. As for their profession, 45.36% were private employees, 31.62% were retired, and the rest belonged to different smaller categories. Finally, 84.09% of the patient sample had Greek nationality and almost all declared Attica as their prefecture of residence, the majority of patients had economic difficulties. The structure of the research sample is presented in Table 1.

### 3.3. Outpatients’ Measurement of Satisfaction and Experiences

Systematic gathering of information on patients’ needs and experiences, using methodologically sound instruments, should be an integral part of routine care. In this study, a structured and validated questionnaire was used, appropriately adjusted to the needs of the study to gather information about outpatients’ reported experiences [31].

### 3.4. Scores of Dimensions Associated with Satisfaction of the Admitted Patients

#### 3.4.1. Outpatient’s Satisfaction with Appointment, Laboratories and Departure

Investigating the profile of the outpatients in relation with their appointment, it appears that 67.89% (Ν = 687) of the total mentioned non-chronic disease, a percentage of 20.26% (Ν = 205) were chronic diseases, 4.55% (Ν = 46) were for drug prescription, 3.46% (Ν = 35) were for emergency care, 3.06% (Ν = 35) were for preventive care, and only 0.79% (Ν = 8) were concerned with the granting of a certificate. The most common reasons for patient visits were pathological (26.48%, N = 268), heart diseases (9.78%, N = 99), orthopedics (9.49%, N = 96), gynecological conditions (8.70%, N = 88), and ophthalmic problems (7.31%, N = 74). Lung-related problems accounted for 7.02% (N = 71) of visits, dental problems for 5.73% (N = 58), dermatological conditions for 5.14% (N = 52), otolaryngological conditions for 4.25% (N = 43), pediatric conditions for 2.96% (N = 30), and general surgery for 2.47% (N = 25). Almost two-thirds of the sample (76.68%, N = 776) arranged their appointment by phone, 13.83% (Ν = 140) via internet, and 9.49% (N = 96) by visiting the health center of their interest. Table 2 shows that only 57.55% (N = 568) expressed satisfaction with the options they had for choosing the day and time of their appointment. A percentage of 57.81% (N = 585) had to wait for an appointment for 16–30 days, 30.63% (N = 310) between 1 and 2 months, 8% (N = 81) less than 15 days, and 3.56% (N = 36) went without an appointment. Importantly, low satisfaction (52.65%, N = 526) was expressed due to the waiting time from the moment the patient wanted to be examined to the point of PHC visit.

The registration staff behavior at the entrance was rated as low (66.93%, N = 674). Hence, it was more difficult to find a seat to wait in the waiting room before the examination (56.68%, N = 573) and it took longer to be examined (47.28%, N = 478). According to the majority of patients (89.91%, N = 909), physicians provided understandable information about their health condition. Concerning the technical skills of the physicians, they were rated satisfactorily by patients (84.76%, N = 854). In the case of treatment, the majority of patients (84.05%, N = 843) indicated that the explanations and instructions received from the physician were appropriate. In medical examination, privacy (89.43%, N = 905) and respect (76.55%, N = 754) were significantly satisfactory dimensions (Table 2). However, the time taken for physician to examine patients (65.80%, N = 658) and listen to them (65.21%, N = 643) did not seem to be enough. Usually, the average consultation with a physician lasts 10–15 min, depending on the reason for the encounter. It takes less than 10 min for patients to request a prescription, and more than 15 min for those with new health problems. In this particular study, only four (0.40%) patients stayed in the medical room for less than 5 min. Most of the samples (593 or 58.60%) stayed between 6 and 10 min, 406 (40.12%) stayed for 11–20 min, 6 (0.59%) between 21 and 30 min, and 3 (0.30%) stayed for more than 31 min. Regarding the nursing staff, satisfaction was expressed in relation to their courtesy (88.31%), and also with the willingness to answer patients’ questions (88.31%). Less than half of all samples (42.39%, N = 429) were users of laboratory services, and the majority of them reported that staff were reluctant to serve them (91.14%), but there were delays in performing laboratory tests (44.29%). While only 51.19% (N = 473) declared that they were satisfied with the check-out procedure at departure, the main reasons for dissatisfaction were bureaucratic procedures and delays, as well as the helpfulness and politeness of the staff (42.39%, N = 429).

#### 3.4.2. Outpatient’s Satisfaction and Sociodemographic Characteristics

Sociodemographic characteristics and health state are principal components in the conceptual model of patient satisfaction in primary care. The influence of sociodemographic characteristics or subjective health status assessment appear to be relatively insignificant compared to the influence of health care service accessibility and actual experiences [38,39]. Our study revealed that men were more satisfied than women. Specifically, men were more satisfied regarding technical skills of physicians (ms = 2.99/5) and behavior of nursing staff (3.17), whereas dissatisfaction was reported regarding proper signboards/direction guides (1.92), infrastructure of facilities (2.01) and their cleanliness (2.02). Women reported satisfaction regarding the behavior of nursing staff and laboratories, whereas dissatisfaction was reported regarding infrastructure of facilities (2.05) and their cleanliness (2.01). Pertaining to this, some studies have shown that women report less satisfaction than men, as they tend to rate care more negatively than men.

According to our findings, the younger age group were more satisfied than the older (Table 3). Notably, patients belonging to the age group of 18–30 years scored higher οn most sub-scales of satisfaction compared to the other age groups. The age group of 31–44 years scored higher about behavior of nursing staff (3.22) and waiting time before the appointment (2.99), and the group of 61–74 years scored higher about technical skills of physician (2.99) and behavior of nursing staff (3.20). Dissatisfaction and low rates belonged to the age group >75 and especially about facilities (2.30) and their cleanliness (2.02), communication of personnel (2.17), humaneness (2.18), arrival and admission (2.27), and contribution of the PHCs (2.14). The married participants were slightly (2.62) more satisfied than singles/divorced/widowed (2.61) persons. According to nationality, Greeks (2.61) were more dissatisfied than foreign patients (2.65). Interestingly, the literate group (2.58) was less satisfied regarding the majority of aspects of care as compared to the illiterate or secondary group. This could be due to education leading to a better understanding and an unmet expectation of higher level of care. 

#### 3.4.3. Outpatient’s Satisfaction and Person-Centered Care

In primary health care, increased attention has been paid to person-centered care, as this type of care embodies more on the uniqueness of a person with an illness or impairment. Many surveys have shown that person-centered care is associated with shorter hospital stays, lower readmission rates, higher quality of care, and enhanced patient satisfaction with health care [9,40,41,42,43,44,45,46,47,48]. It can also lead to increased patient self-efficacy; less anxiety, pain, and depression; and fewer referrals or additional patient research [49,50,51]. Important determinants of patient-centered care are accessibility and availability, communication, humaneness, continuous and comprehensive care, information, coordination, and waiting time. These are all aspects of ‘person-focused’ care, which means that physicians focus on a ‘whole person’ rather than on a ‘patient’, i.e., their health conditions. Distinguishing from other questionnaires, the specific questionnaire in the way it was structured gives important information about the dimensions of human-centered care [31,47,52,53].

Regarding the approach to primary health care facilities, 57.61% (N = 583) of the total sample reported that they used a private vehicle, of which more than half (55.92%) stated that they did not have difficulty finding parking, about one-third (30.04%, N = 304) preferred to go by walking, 10.87% (N = 110) used urban transportation, and only 1.48% (N = 15) arrived by taxi. Obviously, PHC centers are affordable and conveniently located for patients. Conversely, causes of patients’ dissatisfaction were connected with difficulties in choosing the day and time for their appointment as only 56.13% were satisfied, long waiting times from the moment they wanted to be examined to the moment they visited the health center (51,98%), and the inadequacy of seating arrangement in the waiting area (56.62%).

A serious access barrier related to providers’ availability is identified, as out-of-hour services were not provided on a 24/7 basis, more specifically only one PHC center on a pilot basis provided 24/7 services. Another severe barrier was the failure of the reform of the so-called “family doctor”, as was implemented in 2017 but did not achieve success, partly because the low salaries of physicians. Later, the concept of “personal doctor” was legislated in 2022 for enrollment, empanelment, and rostering patients and tailoring their medical profile, but it did not work as only 20% of the population was registered, creating a significant gap in providers’ access to patient medical history [4,21,54,55,56,57]. As such, seamlessnes has not been achieved, since long-term relationships can only be achieved when patients can meet their personal physician and if physicians offer broader service profiles. In this survey, accessibility and availability were rated as weak (2.19/5).

Effective communication and interpersonal skills of employees are achieved when professionals show compassion, have empathy, learn about their patients’ situations by listening and observing, use simple language (avoiding medical jargon), convey tailored and accessible information/materials, check the patient’s understanding of the information and their reactions to it, or deploy an interpreter. Indeed, there are evidences that good interpersonal skills help develop trust in the patient–physician relationship that promotes collaboration, especially with older adults [58,59,60]. Key elements of successful communication are whether the physician provides enough time to examine their patients and if the physician is willing to listen to what the patients themselves have to say about their health; the results of these questions were 65.80% (ms = 2.54) and 63.54% (2.42), respectively. Consultation time is crucial dimension in our survey, as is well highlighted in health literacy [61,62]. It should be noted that patients seeking a simple prescription or urgent care are more likely to be less satisfied with the services provided and disappointed with their interaction with administrative staff. Communication skills rated at 2.18.

Humaneness is another significant dimension consisting of kindness, empathy, concern, sensitivity, and friendliness that the staff demonstrate to patients and their companions. Perceived nonrespectful attitude and unapproachable interaction style of formal health care providers were reported as barriers to formal health care utilization. This, in turn, can trigger a spiral of information searches, second opinions, and in some cases even obtaining a desired treatment privately [63,64]. The overall humaneness was 80.08% (ms = 2.25), while politeness of the personell at check-in was 66.93%, at the check-out desk was 76.19%, and physicians’ respect for patients was 76.55%.

Greece is among the countries showing the least development of digital technologies in the provision and management of health care within the European Union. Proper development of interoperable e-health applications, the integration of telehealth, remote or home monitoring, electronic health event summaries, laboratory results access, and digital health information (electronic health records-EHR) to record minimal morbidity data are basic deficiencies of the system. In a similar vein, PHC centers rarely offer dedicated clinical sessions, and only occasional telephone or online consultations took place. In these conditions, information transfer is insufficient and professionals have difficulties in monitoring their patients’ interactions across different health care settings [65,66,67,68].

The previous set of evidence shows that there are barriers to long-term relationships between primary care and patients. Person-centered care is rated as weak (2.49 ± 0.28) and does not meet patients’ medical needs; as a result, one- to two-thirds of patients attend hospital emergency departments with problems that could be managed at PHC level or which led to out-of-pocket spending [69,70,71].

#### 3.4.4. Overall Outpatient’s Satisfaction

According to the results, patient satisfaction depends on complex factors related to the patient’s needs and it appears that these are directly related to the effectiveness of the services are provided, as well as their facilities. In particular, the overall mean satisfaction with PHC services was moderate (2.62 ± 0.18), thus demonstrating that the impression of the quality of health services was mediocre. Neutral scores were found regarding behavior of nursing staff (3.19 ± 0.29), laboratories (2.99 ± 0.29), waiting before the appointment (2.98 ± 0.26), and departure (2.55 ± 0.44), while the least satisfaction was observed regarding the cleanliness of toilets (2.20 ± 0.71), the contribution of the PHCs (2.29 ± 0.52), arrival and admission (2.31 ± 0.40), and the medical examination and behavior of physician (2.36 ± 0.35).

#### 3.4.5. Multiple Linear Regression Analysis of Predictors Associated with Outpatient Satisfaction

Exploratory factor analysis was used to classify the questionnaire and define three dimensions, namely, (1) medical staff, (2) nursing staff, and (3) appointment time [31]. Moreover, the multiple linear regression analysis was employed to identify the determinants that have the most significant impact on patient satisfaction in public primary health care centers. From the result of the coefficient in Table 4, it can be understood that “Medical staff” with (β = 0.427; *p* < 0.001) and “Appointment time” with (β = 0.390; *p* < 0.001) are the strongest predictors of outpatient satisfaction, followed by “Nursing staff” (β = 0.019; *p* = 0.543). Furthermore, using Bonferroni correction, patient satisfaction is associated with health care services with (F = 151.52, *p* < 0.05).

In the linear regression, it was possible to determine that regarding the time aspect (time consultation), men (*p* = 0.04 < 0.05) were more satisfied than women. Furthermore, older patients (β = 0.183, *p* < 0.05) were more satisfied than their younger counterparts. However, older patients (β = −0.201, *p* < 0.05) were more dissatisfied than younger patients regarding the time consultation, meaning that their expectation was not met (<0: negative range of the target–actual comparison). Married participants were more satisfied regarding the time consultation (*p* = 0.03) and physicians (*p* = 0.00) compared to the rest. In addition, educated patients (β = −0.095, *p* < 0.05, <0: negative range of the target-actual comparison) were more strict and less satisfied in relation with those who had basic education.

#### 3.4.6. Reasons Related to Preferring, Recommending, and Generally Favorable Thoughts about the Services Provided in PHC Centers

Table 5 shows the main influencing factors that drive patients to prefer and recommend a PHC center and generally have positive feelings and thoughts about the services provided.

Multivariate linear regression revealed that medical examination and behavior of physician, communication and technical skills of physician, behavior of nursing staff, waiting time for an appointment, arrival and admission, laboratories, infrastructure of facilities, and the general contribution of the PHC center in community were the key influencing factors (*p* ≤ 0.05) related to preferring and recommending a PHC center. According to the scores that the twenty PHC centers received on a scale of zero to ten, from the patients who visited them for their personal health, the mean score was 6.75 (or 67.50%). Notably, the PHC center which was rated first with 7.19 (or 71.90%) was the first urban PHC center in Greece with an emergency department working 24 h a day, located in the center of Athens, near the all means of transportation [72].

## 4. Discussion

The demand for a robust primary health care is evolving rapidly in European countries in a context of population growth, prolonged life expectancy, climate change, and a variety of health problems combine to increase the demand for medical services. This increase is challenging, as patients with chronic disease multimorbidity have complex care needs that often make it difficult and expensive to manage adequate health care delivery, while at the same time increasing the need for patients to stay longer in their own private spaces [73]. Similarly, the patient’s condition needs to be well managed at the primary care level and greater person-centered care is needed to live independently and have a high life expectancy [74]. In 2012, a population survey was conducted among 3,020 users of primary care services in seven European countries (Estonia, Lithuania, Hungary, Finland, Spain, Italy, and Germany), which showed us that German (57.21%) and Lithuanian patients (65.81%) were less satisfied with primary care services compared with those from Finland (74.07%), Spain (78.89%), and Estonia (81.16%), whereas Hungarian (85.31%) and Italian patients (86.74%) were more satisfied.

Patient satisfaction depends on a number of complex determinants, including patient characteristics and primary care provider-related factors. The factors of quality services considered most important by European outpatients in Bulgaria, Hungary, Lithuania, and Romania were physician’s skills and modern medical equipment, followed by waiting time, examining time, and proximity [75]. In Poland, patients were at most satisfied with the accessibility and equity of care and less satisfied with the coordination and comprehensiveness. However, in other countries, communication, patient-centeredness, coordination, and continuity of care were rated as more important than items related to access [76]. Patients of empathic physicians experience better enablement and also report higher satisfaction with the services. Similarly, longer patient–physician relationship was found as the most influential determinants of higher satisfaction in countries such as the Netherlands, Norway, and Finland, where more than 95% of people have a primary care clinician. According to national estimates, when primary care fails to meet patient needs, “inappropriate” visits to hospital emergency departments are made; 20% in Italy, 31% in Portugal, and 56% in Belgium [77,78,79]. Considering the above, differences in organizational, financial, regulatory, cultural, and financial characteristics of a health system affect the structure of primary care, physician behavior, and professional–patient relationships.

Health care services are structured differently across the countries of European Union, as the initial prerequisites, reasons, and methods for implementing reforms all differ. Ιn Beveridge-type countries with a strong general practice and gatekeeping role, such as Finland, the Netherlands, Hungary, Ireland, Italy, Lithuania, and Spain, these have been shown to be associated with appropriate referral for specialist care or hospitalization, lower costs, and achieving better health care outcomes [80,81]. Conversely, in countries with the Bismarck system such as Austria, the Czech Republic, Luxembourg, Cyprus, and Greece, patients have direct access to most physicians and secondary care. In other countries, such as France, Denmark, Norway, and Sweden, patients have also direct access to secondary care without referral, but there are financial incentives to register with a primary care physician to encourage greater coordination and continuity of care. Upon mapping the European Union based on three criteria (Figure 1)—(a) the level of primary care, (b) the type of health care system, and (c) life expectancy at birth—we found that countries that adopt a gatekeeping system have stronger primary care and achieve higher levels of life expectancy [82,83].

In Greece, the level of primary care is low compared to other European Union countries (Figure 1), especially since there are severe drawbacks affecting the health system, specifically the following:

(a) The economic crisis between 2009 and 2019 affected the overall performance and staffing of primary health care. Declining health care spending has reduced the quality of care; thus, skilled professionals are more likely to seek better advancement opportunities in the private sector or in other countries that provide them with better payments, training, and integrated systems for their work. Research has shown that longitudinal, relationship-based primary care with a single physician is associated with better outcomes and continuity.

(b) The lack of a long-term strategy and consensus on the role of primary care in the health system. PHC centers do not provide health care services 24 hours, 7 days a week. However, as off-hours incidents and timely access to care has become a common concern in this age of aging population, this is a compelling issue. Similarly, there is no gatekeeping system, which leads to a large accumulation of patients in hospital’s emergency departments. In our survey, the PHC center which was rated first by outpatients was the first urban PHC center in Greece that for the first time started an emergency department working 24 h a day.

(c) The lack of digitalization. Greece has the least progress within the European Union. Unfortunately, at present, adoption of the electronic health record (EHR) has not materialized, and this has coincidences for safety, timeliness, continuity, effectiveness, efficiency, equity, and patient-centeredness. Hence, lack of efficient information flow (between primary and secondary care) could be another explanation of worse evaluation of coordination of care. In 2020, the country increased the use of e-prescriptions, which are sent directly to patients by SMS or email, so that patients do not have to leave their homes, a beneficial solution for the patient and the system, which was, however, scheduled for implementation before 2012.

(d) To provide patient-centered care, it is necessary to know what patients prefer and not make assumptions about their preferences. Τhe lack of patient associations and federations that participate in the dialogue and decision-making processes for health policies is a disadvantage [84].

(e) The massive influx of immigrants into the country resulted in straining an already fragmented system and at the same time led to the transmission of certain respiratory or intestinal pathogens and the emergence of many infectious diseases such as tuberculosis, hepatitis, measles, rubella, etc.

Under these circumstances, citizens in Greece show greater trust in hospital care and continue to increase private spending and out-of-pocket payments for physicians and laboratory services to maintain a decent level of health. Similar are the results of other Greek surveys, which recorded weak to moderate outpatient satisfaction [21,85,86,87,88,89,90,91,92].

## 5. Conclusions

Health should be seen as an investment in our society, our economy and our future. The outbreak of the COVID-19-pandemic revealed the fragility of health systems in the European Union, as Central and Eastern European countries were particularly affected, after experiencing the largest decline in life expectancy since World War II. To address the challenges of these demographic and epidemiological changes, the EU health systems need to continuously strengthen the primary health systems to provide comprehensive and coordinated care for their populations. Even if large variations exist in the organization of primary care between European countries, member states’ best practices should inspire other members and be adopted.

The overarching goal of this research study was to delineate the level of patients’ satisfaction in Greece and gauge health care quality from the patient’s viewpoint. In our study, οverall satisfaction was rated at a moderate level (2.62), while person-centered care as weak (2.49). Based on the results of the statistical analysis, it was found that “Medical staff” (β = 0.427; *p* < 0.001) and “Appointment time” (β = 0.390; *p* < 0.001) were the strongest predictors of outpatient satisfaction, followed by “Nursing staff” (β = 0.019; *p* = 0.543). According to the scores that the twenty PHC centers received from the patients who visited them, the mean score was 6.75/10 (or 67.50%). A strength of this study is the application of qualitative methodologies to elicit rich descriptive data to provide further context and gain a better understanding of the key influencing factors (*p* ≤ 0.05) that drive patients to prefer and recommend a PHC center, such as medical examination, skills of physician, consultation time, waiting time for an appointment, accessibility, laboratories, infrastructure of facilities and the general contribution of the PHC center in community. Nursing professionals can take on a more prominent role in primary care through multidisciplinary teams. Aligning the provision of appointments should be more coordinated with the utilization of digital health applications and telehealth consultations for those who did not need face-to-face consultations. The computerization of the patient medical records in Greece will lead to improved access to patients’ up-to-date medical history, investigation results, and previous consultations and hospitalizations. When primary care is well equipped to manage most mild cases, pressure is relieved on secondary or tertiary care. Making care more patient-centered may be the way forward. Consequently, Greece must proceed with comprehensive reforms to strengthen the primary care system to better address the patients’ needs.

As with any study, this research has a few limitations. The findings of the survey referred to twenty out of fifty-seven of total public PHC centers in the capital of Athens. Therefore, it is not possible to draw firm conclusions about the whole population of Greek outpatients. Moreover, this survey was fully based on public urban PHC centers; private or rural PHC centers were not included here.

## Figures and Tables

**Figure 1 healthcare-12-01427-f001:**
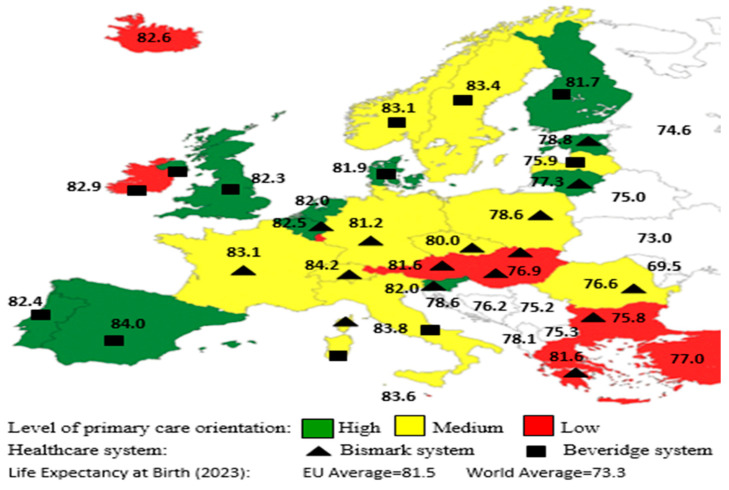
Level of primary care orientation and life expectancy at birth by type of health care system in Europe.

**Table 1 healthcare-12-01427-t001:** Sociodemographic characteristics of outpatients (Ν = 1012).

Characteristics	Frequency	Percentage
Gender		
Male	383	37.85%
Female	629	62.15%
Age (years)		
18–30	70	6.91%
31–44	166	16.40%
45–60	394	38.93%
61–74	357	35.28%
75>	25	2.47%
Marital status		
Married	730	72.13%
Single	143	14.13%
Divorced	53	5.24%
Widowed	86	8.50%
Education level		
Illiterate	12	1.19%
Primary school	150	14.82%
Secondary school	223	22.04%
Compulsory	534	52.77%
University	76	7.51%
Master or PhD	17	1.68%
Profession		
Civil servant	47	4.64%
Private employee	459	45.36%
Retired	320	31.62%
Worker	15	1.48%
Student	30	2.96%
Freelancer	27	2.67%
Unemployed or domestic	98	9.68%
Other	16	1.58%
Economic situation		
I cannot cope with my financial obligations	5	0.49%
I manage financially with great difficulties	894	88.34%
I manage financially but I do not have much left aside	111	10.97%
I do not know or I do not answer	2	0.20%
Nationality		
Greek	851	84.09%
Other	161	15.91%
Prefecture of residence		
Attica	1011	99.90%
Other	1	0.10%

**Table 2 healthcare-12-01427-t002:** Descriptive statistics of the admitted outpatients (Ν = 1012).

		Positive Reports *										
	Questions (Items)/Dimensions	N	%	Mean Scores (MS)	SE **	Median	Variance	SD **	Min	Max	Range	IQR **
Q1	Your appointment has been made easy and fast	761	75.20%	2.35	0.029	2.00	0.845	0.919	1	5	4	0
Q2	It was difficult to choose day and appointment time	568	56.13%	2.67	0.035	2.00	1.218	1.104	1	5	4	1
Q3	Your access to the Health Center was easy	990	97.83%	1.94	0.012	2.00	0.149	0.386	1	4	3	0
Q4	The staff at the information desk were polite and eager to serve you	674	66.60%	2.38	0.022	2.00	0.491	0.701	1	5	4	1
Q5	The waiting time from the moment you wanted to be examined to the moment you visited the Health Center it was big	526	51.98%	2.77	0.029	2.00	0.864	0.930	1	5	4	2
Q6	You are happy with the accessibility for people with disabilities	896	88.54%	2.11	0.018	2.00	0.329	0.573	1	5	4	0
Q7	The signage in the premises of the Health Center for the dispensaries was incomplete	941	92.98%	1.98	0.019	2.00	0.350	0.592	1	5	4	0
	Arrival and admission: Q 1–7 (7)	765	75.61%	2.31	0.013	2.29	0.161	0.402	1	3.57	2.57	0.57
Q12	The waiting room was clean	990	97.83%	1.99	0.009	2.00	0.086	0.293	1	5	4	0
Q13	The waiting room was cramped	923	91.21%	2.06	0.018	2.00	0.317	0.563	1	5	4	0
Q14	It was easy to find a seat to wait	573	56.62%	2.84	0.032	4.00	1.019	1.010	1	5	4	2
Q15	The temperature of the waiting room was unpleasant (it was too cold or too hot)	964	95.26%	1.85	0.019	2.00	0.382	0.618	1	5	4	0
Q16	The time you waited in the waiting room was too long	478	47.23%	2.96	0.032	2.00	1.026	1.013	1	5	4	2
	Waiting before the appointment: Q 12–16 (5)	786	77.63%	2.98	0.008	2.90	0.070	0.265	2.10	3.90	1.80	0.20
Q18	The toilets were clean and well maintained	269	87.62%	2.20	0.041	2.00	0.504	0.710	1	5	4	0
Q19	The examination room was clean	979	96.74%	2.00	0.010	2.00	0.099	0.314	1	4	3	0
Q20	The examination room was cramped and uncomfortable	869	85.87%	2.13	0.018	2.00	0.335	0.579	1	4	3	0
Q21	The physician took enough time to examine you	658	65.80%	2.54	0.031	2.00	0.942	0.970	1	5	4	1
Q22	The physician showed disregard for proper seclusion during the examination so that you are not seen or heard by people who should not be	905	89.43%	1.90	0.020	2.00	0.397	0.630	1	5	4	1
Q23	The physician was willing to listen to what you had to say about your health	643	63.54%	2.42	0.026	2.00	0.688	0.830	1	5	4	1
Q24	Your physician has given you insufficient or incomprehensible information regarding your health condition	909	89.82%	1.96	0.020	2.00	0.411	0.641	1	5	4	0
Q25	You have confidence in the correctness and appropriateness of the diagnosis and treatment given to you by the physician	682	67.39%	2.40	0.029	2.00	0.835	0.914	1	5	4	1
Q26	Were the explanations and instructions you received from the doctor regarding the treatment (how to take medicine, how to recover, etc.) inadequate or confusing?	843	83.30%	2.11	0.023	2.00	0.548	0.741	1	5	4	0
Q27	Your physician treated you with respect	754	74.51%	2.26	0.024	2.00	0.548	0.740	1	5	4	0
Q28	The physicians seemed incompetent and poorly trained	865	85.47%	3.82	0.020	4.00	0.410	0.640	1	5	4	0
	Medical examination and behavior of physician: Q 19–28 (10)	811	80.11%	2.36	0.011	2.30	0.120	0.346	1.50	3.50	2	0.38
Q30	The nursing staff was kind to you	589	88.31%	2.10	0.016	2.00	0.171	0.413	1	5	4	0
Q31	The nursing staff was reluctant to answer your questions	566	84.86%	2.05	0.022	2.00	0.330	0.574	1	5	4	0
	Behavior of nursing staff: Q 30–31 (2)	578	86.58%	3.19	0.009	2.00	0.083	0.288	1	5	4	0
Q33	The waiting time for laboratory tests was long	190	44.29%	3.03	0.048	2.00	0.999	1.000	1	4	3	2
Q34	Signposting in the Health Center grounds to find the labs was adequate	385	89.74%	2.17	0.033	2.00	0.464	0.682	1	5	4	0
Q35	The examination areas in the laboratories were cramped/uncomfortable	359	83.68%	2.23	0.033	2.00	0.476	0.690	1	5	4	0
Q36	The laboratory tests were carried out in an easy and painless way for you	382	89.04%	2.14	0.028	2.00	0.337	0.581	1	4	3	0
Q37	The labs support staff were reluctant to serve you	391	91.14%	2.08	0.027	2.00	0.318	0.564	1	5	4	0
	Labotarories: Q 33–37 (5)	341	79.58%	2.99	0.014	3.20	0.085	0.292	2	3.80	1.80	0.60
Q39	Payment/visa document procedures at check-out were time consuming	473	46.74%	2.95	0.033	2.00	0.996	0.998	1	5	4	2
Q40	The secretariat staff at check-out were polite and helpful	704	69.57%	2.27	0.018	2.00	0.286	0.535	1	4	3	0
Q41	Are you satisfied with the way your companions or relatives are treated by the Secretariat staff at check-out?	537	53.06%	2.20	0.019	2.00	0.236	0.486	1	5	4	0
	Departure: Q 39–41 (3)	571	56.46%	2.55	0.014	2.67	0.190	0.436	1	4	3	0.34
Q42	Are you satisfied with the contribution of the Health Center’s health services to improving your health status?	799	78.95%	2.27	0.021	2.00	0.452	0.672	1	5	4	0
Q43	Are you satisfied with the quietness of the Health Center’s?	832	82.21%	2.30	0.023	2.00	0.534	0.731	1	5	4	0
	Contribution of the PHCs: Q 42–43 (2)	816	80.58%	2.29	0.016	2.00	0.265	0.515	1	4.50	3.50	0
	Overall Satisfaction: Q 1–7, 12–16, 18, 19–28, 30–31, 33–37, 39–41, 42–43 (35)	617	78.02%	2.62	0.006	2.59	0.032	0.180	1.80	3.26	1.46	0.11
	Accessibility and Availability: Q 3, 6, 21 (3)	848	83.80%	2.19	0.013	2.00	0.169	0.411	1	4.33	3.33	0.67
	Technical skills: Q 26, 28 (2)	854	84.39%	2.97	0.011	3.00	0.126	0.356	2	5	3	0
	Communication: Q 23, 24, 31 (3)	706	79.41%	2.18	0.017	2.00	0.287	0.536	1	4	3	0.33
	Humaneness: Q 4, 27, 30, 37, 40–41 (6)	608	73.86%	2.25	0.011	2.17	0.336	0.336	1	4	3	0.33
	Waiting times: Q 5, 16, 33, 39 (4)	417	47.56%	2.89	0.020	2.50	0.419	0.648	1	5	4	1.13
	Infrastructure of facilities: Q 13, 15, 20, 35 (4)	779	89.00%	2.04	0.013	2.00	0.165	0.406	1	5	4	0.25
	Cleanliness of the facilities: Q 12, 19 (2)	985	97.28%	2.03	0.011	2.00	0.128	0.358	1	4	3	0
	Proper signboards/direction guides: Q 7, 34 (2)	663	91.36%	2.02	0.017	2.00	0.283	0.532	1	5	4	0.75

Notes: * In the two columns, the number and percentage of patients strongly agreeing or merely agreeing with a positive statement in each questionnaire item are presented; ** SE = Standard Error; SD = Standard Deviation; IQR = Interquartile Range; *** Negative = 0–2.49; Neutral = 2.50–3.49; Positive = 3.50–5.

**Table 3 healthcare-12-01427-t003:** Dimensions of satisfaction and association with various sociodemographic variables (N = 1012).

			Gender			Age (Years)						Marital Status			Literacy Status				Nationality			Total
			(Mean ± SD)		Unpaired *t*-Test	(Mean ± SD)					ANOVA	(Mean ± SD)		Unpaired *t*-Test	(Mean ± SD)			ANOVA	(Mean ± SD)		Unpaired *t*-Test	(Mean ± SD)
Dimensions	Questions	Items	Male	Female	(df = 1011)	18–30	31–44	45–60	61–74	>75	F(P)	Married	Other	(df = 1010)	Illiterate, Primary	Secondary	University, Master, PhD	F(P)	Greece	Other	(df = 1010)	
Arrival and admission	Q 1–7	7	2.27 ± 0.43	2.33 ± 0.38	742.335 (0.03)	2.37 ± 0.43	2.31 ± 0.40	2.31 ± 0.41	2.30 ± 0.40	2.27 ± 0.32	0.473 (0.76)	2.30 ± 0.41	2.32 ± 0.39	528.208 (0.47)	2.30 ± 0.41	2.34 ± 0.38	2.17 ± 0.48	4.208 (0.00)	2.31 ± 0.40	2.35 ± 0.39	230.168 (0.20)	2.31 ± 0.40
Waiting before the appointment	Q 12–16	5	2.99 ± 0.27	2.97 ± 0.26	790.249 (0.22)	2.99 ± 0.36	2.99 ± 0.23	2.97 ± 0.26	2.99 ± 0.25	2.85 ± 0.34	1.890 (0.11)	2.97 ± 0.24	2.99 ± 0.32	412.095 (0.49)	3.02 ± 0.27	2.96 ± 0.26	2.90 ± 0.25	6.964 (0.00)	2.97 ± 0.27	3.01 ± 0.24	243.047 (0.10)	2.98 ± 0.26
Cleanliness of toilets (b)	Q 18	1	2.18 ± 0.76	2.20 ± 0.68	230.677 (0.84)	2.83 ± 1.40	2.22 ± 0.71	2.26 ± 0.78	2.08 ± 0.49	1.91 ± 0.30	4.195 (0.00)	2.20 ± 0.69	2.18 ± 0.77	130.723 (0.85)	2.08 ± 0.53	2.13 ± 0.63	2.70 ± 1.09	9.370 (0.00)	2.19 ± 0.72	2.22 ± 0.64	47.696 (0.81)	2.20 ± 0.71
Medical examination and behavior of phycisian	Q 19–28	10	2.36 ± 0.34	2.35 ± 0.35	835.975 (0.67)	2.41 ± 0.37	2.35 ± 0.36	2.33 ± 0.36	2.38 ± 0.32	2.39 ± 0.33	1.490 (0.20)	2.34 ± 0.32	2.42 ± 0.39	437.952 (0.00)	2.43 ± 0.35	2.33 ± 0.33	2.23 ± 0.33	8.973 (0.00)	2.35 ± 0.35	2.39 ± 0.34	226.007 (0.15)	2.36 ± 0.35
Behavior of nursing staff	Q 30–31	2	3.17 ± 0.27	3.21 ± 0.29	851.961 (0.02)	3.11 ± 0.27	3.22 ± 0.27	3.19 ± 0.30	3.20 ± 0.29	3.13 ± 0.28	2.101 (0.08)	3.20 ± 0.29	3.16 ± 0.27	552.388 (0.03)	3.19 ± 0.30	3.20 ± 0.28	3.16 ± 0.26	0.573 (0.72)	3.19 ± 0.29	3.19 ± 0.28	228.285 (0.88)	3.19 ± 0.29
Labotarories (c)	Q 33–37	5	2.96 ± 0.27	3.01 ± 0.30	353.738 (0.07)	3.04 ± 0.28	2.91 ± 0.28	3.00 ± 0.31	3.03 ± 0.28	2.96 ± 0.27	2.397 (0.05)	3.01 ± 0.29	2.94 ± 0.28	353.738 (0.03)	3.02 ± 0.25	2.96 ± 0.30	3.08 ± 0.35	2.428 (0.05)	3.00 ± 0.30	2.97 ± 0.24	115.412 (0.38)	2.99 ± 0.29
Departure	Q 39–41	3	2.57 ± 0.47	2.54 ± 0.41	711.404 (0.34)	2.52 ± 0.36	2.51 ± 0.46	2.58 ± 0.46	2.54 ± 0.40	2.57 ± 0.44	0.731 (0.57)	2.56 ± 0.44	2.53 ± 0.42	436.011 (0.33)	2.53 ± 0.42	2.56 ± 0.44	2.61 ± 0.49	1.327 (0.25)	2.56 ± 0.42	2.52 ± 0.49	205.877 (0.39)	2.55 ± 0.44
Contribution of the PHCs	Q 42–43	2	2.35 ± 0.54	2.25 ± 0.49	745.483 (0.01)	2.31 ± 0.54	2.33 ± 0.56	2.28 ± 0.52	2.28 ± 0.50	2.14 ± 0.34	0.815 (0.52)	2.29 ± 0.50	2.29 ± 0.56	462.926 (0.99)	2.34 ± 0.53	2.28 ± 0.50	2.12 ± 0.53	3.029 (0.01)	2.26 ± 0.50	2.43 ± 0.55	214.598 (0.00)	2.29 ± 0.52
Accessibility and Availability	Q 3, 6, 21	3	2.17 ± 0.42	2.20 ± 0.40	780.119 (0.31)	2.19 ± 0.57	2.18 ± 0.38	2.20 ± 0.43	2.18 ± 0.37	2.23 ± 0.37	0.232 (0.92)	2.19 ± 0.38	2.19 ± 0.47	432.166 (0.85)	2.22 ± 0.39	2.17 ± 0.38	2.20 ± 0.61	2.748 (0.02)	2.18 ± 0.41	2.23 ± 0.42	219.255 (0.17)	2.19 ± 0.41
Technical skills	Q 26, 28	2	2.99 ± 0.33	2.96 ± 0.37	879.062 (0.18)	2.98 ± 0.49	2.97 ± 0.27	2.95 ± 0.39	2.99 ± 0.33	2.96 ± 0.14	0.607 (0.66)	2.95 ± 0.33	3.03 ± 0.40	436.554 (0.01)	2.99 ± 0.36	2.95 ± 0.35	3.01 ± 0.38	1.867 (0.10)	2.98 ± 0.37	2.93 ± 0.29	266.745 (0.08)	2.97 ± 0.36
Communication	Q 23, 24, 31	3	2.18 ± 0.51	2.17 ± 0.55	854.039 (0.67)	2.12 ± 0.61	2.13 ± 0.56	2.14 ± 0.56	2.25 ± 0.47	2.17 ± 0.51	2.556 (0.04)	2.17 ± 0.51	2.20 ± 0.59	454.759 (0.49)	2.29 ± 0.53	2.12 ± 0.52	2.00 ± 0.53	7.742 (0.00)	2.17 ± 0.53	2.20 ± 0.55	454.759 (0.49)	2.18 ± 0.54
Humaneness	Q 4, 27, 30, 37, 40–41	6	2.27 ± 0.34	2.23 ± 0.33	782.044 (0.10)	2.30 ± 0.49	2.24 ± 0.34	2.22 ± 0.35	2.28 ± 0.28	2.18 ± 0.30	2.380 (0.05)	2.24 ± 0.31	2.28 ± 0.40	412.447 (0.10)	2.30 ± 0.31	2.25 ± 0.32	2.01 ± 0.41	13.021 (0.00)	2.24 ± 0.34	2.31 ± 0.30	247.409 (0.01)	2.25 ± 0.34
Waiting times	Q 5, 16, 33, 39	4	2.94 ± 0.61	2.85 ± 0.67	862.473 (0.03)	2.80 ± 0.70	2.98 ± 0.61	2.80 ± 0.65	2.98 ± 0.63	2.60 ± 0.56	6.334 (0.00)	2.87 ± 0.63	2.92 ± 0.70	462.326 (0.33)	3.03 ± 0.59	2.83 ± 0.62	2.60 ± 0.85	11.412 (0.00)	2.86 ± 0.66	3.00 ± 0.59	240.712 (0.02)	2.89 ± 0.65
Infrastructure of facilities	Q 13, 15, 20, 35	4	2.01 ± 0.35	2.05 ± 0.43	929.987 (0.11)	2.11 ± 0.68	2.02 ± 0.45	2.07 ± 0.41	1.98 ± 0.27	2.30 ± 0.55	5.681 (0.00)	2.02 ± 0.32	2.09 ± 0.56	354.888 (0.04)	1.96 ± 0.29	2.08 ± 0.43	2.11 ± 0.59	6.219 (0.00)	2.04 ± 0.42	2.03 ± 0.35	253.733 (0.76)	2.04 ± 0.41
Cleanliness of the Facilities	Q 12, 19	2	2.00 ± 0.24	1.99 ± 0.25	890.143 (0.63)	2.03 ± 0.24	2.00 ± 0.21	2.01 ± 0.26	1.97 ± 0.25	2.02 ± 0.27	3.645 (0.01)	2.00 ± 0.25	2.00 ± 0.23	415.362 (0.78)	1.96 ± 0.23	2.02 ± 0.25	1.94 ± 0.25	4.588 (0.00)	1.99 ± 0.25	2.02 ± 0.22	345.851 (0.79)	2.00 ± 0.25
Proper signboards/direction guides	Q 7, 34	2	1.92 ± 0.47	2.09 ± 0.56	909.809 (0.00)	2.29 ± 0.75	2.00 ± 0.47	2.07 ± 0.56	1.94 ± 0.45	1.94 ± 0.55	7.831 (0.00)	1.99 ± 0.50	2.12 ± 0.60	436.843 (0.00)	1.92 ± 0.43	2.06 ± 0.55	2.25 ± 0.70	7.788 (0.00)	2.03 ± 0.54	2.00 ± 0.46	252.383 (0.53)	2.02 ± 0.53
Person-centered care	Q 3–6, 16, 21, 23, 24, 26–28, 30, 31, 33, 37, 39–41	18	2.51 ± 0.27	2.48 ± 0.28	840.470 (0.10)	2.48 ± 0.33	2.50 ± 0.27	2.46 ± 0.30	2.54 ± 0.24	2.43 ± 0.25	4.255 (0.00)	2.48 ± 0.25	2.52 ± 0.34	405.218 (0.07)	2.57 ± 0.34	2.47 ± 0.25	2.37 ± 0.25	14.021 (0.00)	2.49 ± 0.28	2.53 ± 0.27	225.795 (0.01)	2.49 ± 0.28
Overall Satisfaction	Q 1–7, 12–16, 18, 19–28, 30–31, 33–37, 39–41, 42–43	35	2.62 ± 0.18	2.61 ± 0.18	787.878 (0.69)	2.64 ± 0.22	2.63 ± 0.17	2.61 ± 0.20	2.62 ± 0.16	2.55 ± 0.11	1.582 (0.18)	2.62 ± 0.16	2.61 ± 0.22	406.108 (0.65)	2.63 ± 0.16	2.61 ± 0.18	2.58 ± 0.23	1.865 (0.10)	2.61 ± 0.18	2.65 ± 0.17	236.043 (0.01)	2.62 ± 0.18

Notes: (a) Negative = 0–2.49; Neutral = 2.50–3.49; Positive = 3.50–5, (b) df = 305, (c) df = 408, (d) The significance of the means was calculated using the Unpaired *t* test and ANOVA, (e) SD = Standard deviation.

**Table 4 healthcare-12-01427-t004:** Multiple linear regression on dimensions of service quality for outpatient satisfaction.

			Patient Satisfaction					CI (95% Confidence Interval)	
Dimensions	Questions	Items	B	SE	β	*t* Value	*p*	Lower Bound	Upper Bound
Medical staff	Q 22–27	6	0.124	0.009	0.427	13.646	<0.001	0.106	0.142
Nursing staff	Q 30 + 31	2	0.007	0.012	0.019	0.608	0.543	−0.016	0.142
Appointment time	Q 1 + 5	2	0.086	0.007	0.390	12.703	<0.001	0.072	0.099
Constant		10	2.099	0.032		66.233	<0.001	2.037	2.161

Notes: Dependent variable = Patient Satisfaction; R^2^ = 0.40; Adjusted R^2^ = 0.41; SE = 0.13; F-value = 151.52; *p* = 0.00, *p* ≤ 0.05.

**Table 5 healthcare-12-01427-t005:** Regression analysis results of the key influencing factors related to preferring and recommending a PHC center and generally having favorable thoughts about the services provided.

	Outpatient Satisfaction					CI (95% Confidence Interval)	
Dimensions	B	SE	β	*t* Value	*p*	Lower Bound	Upper Bound
**Reasons for preferring the same PHC center again**							
Arrival and admission	0.106	0.068	0.080	1.559	0.120	−0.028	0.239
Waiting before the appointment	−0.173	0.146	−0.086	−1.186	0.236	−0.459	0.114
Medical examination and behavior of phycisian	0.698	0.148	0.453	4.732	0.000	0.408	0.988
Behavior of nursing staff	−0.173	0.094	−0.093	−1.844	0.066	−0.357	0.011
Labotarories	−0.150	0.105	−0.082	−1.438	0.151	−0.356	0.055
Departure	0.018	0.064	0.015	0.290	0.772	−0.107	0.144
Contribution of the PHCs	0.132	0.043	0.127	3.033	0.003	0.046	0.217
Accessibility and Availability	0.115	0.071	0.088	1.619	0.106	−0.025	0.255
Technical skills	−0.173	0.083	−0.115	−2.095	0.037	−0.336	−0.011
Communication	0.060	0.072	0.060	0.833	0.405	−0.081	0.200
Humaneness	0.114	0.087	0.071	1.306	0.192	−0.057	0.284
Waiting times	−0.016	0.055	−0.019	−0.291	0.771	−0.125	0.093
Infrastucture of facilities	−0.187	0.086	−0.142	−2.175	0.030	−0.356	−0.018
Cleanliness of the facilities	0.071	0.083	0.038	0.859	0.391	−0.092	0.235
Proper signboards/direction guides	0.074	0.049	0.073	1.507	0.133	−0.022	0.169
**Reasons for recommending the same PHC center again**							
Arrival and admission	0.178	0.078	0.113	2.267	0.024	0.024	0.332
Waiting before the appointment	−0.009	0.168	−0.004	−0.051	0.959	−0.339	0.322
Medical examination and behavior of physician	0.601	0.170	0.330	3.529	0.000	0.266	0.936
Behavior of nursing staff	−0.164	0.108	−0.075	−1.516	0.130	−0.377	0.049
Labotarories	−0.324	0.121	−0.150	−2.684	0.008	−0.561	−0.087
Departure	0.005	0.074	0.004	0.073	0.942	−0.139	0.150
Contribution of the PHCs	0.134	0.050	0.109	2.672	0.008	0.035	0.233
Accessibility and Availability	0.077	0.082	0.050	0.937	0.349	−0.084	0.238
Technical skills	−0.100	0.095	−0.056	−1.051	0.294	−0.288	0.087
Communication	0.175	0.083	0.149	2.121	0.035	0.013	0.338
Humaneness	0.146	0.100	0.077	1.450	0.148	−0.052	0.343
Waiting times	−0.012	0.064	−0.013	−0.191	0.849	−0.138	0.113
Infrastucture of facilities	−0.191	0.099	−0.123	−1.926	0.055	−0.386	0.004
Cleanliness of the facilities	0.143	0.096	0.064	1.491	0.137	−0.046	0.331
Proper signboards/direction guides	0.070	0.056	0.059	1.250	0.212	−0.040	0.181
**Favorable acceptance of the health care services provided ***							
Arrival and admission	−0.125	0.187	−0.056	−0.671	0.502	−0.492	0.242
Waiting before the appointment	0.672	0.295	0.186	2.276	0.023	0.091	1.253
Medical examination and behavior of physician	−0.243	0.334	−0.087	−0.728	0.467	−0.901	0.414
Behavior of nursing staff	0.258	0.198	0.085	1.304	0.193	−0.131	0.646
Labotarories	0.832	0.222	0.270	3.755	0.000	0.396	1.268
Departure	0.214	0.170	0.100	1.258	0.209	−0.120	0.548
Contribution of the PHCs	−0.015	0.143	−0.008	−0.106	0.916	−0.295	0.265
Accessibility and Availability	−0.088	0.151	−0.038	−0.582	0.561	−0.385	0.209
Technical skills	0.248	0.156	0.100	1.592	0.112	−0.058	0.554
Communication	−0.286	0.165	−0.151	−1.730	0.084	−0.611	0.039
Humaneness	0.021	0.207	0.006	0.100	0.920	−0.386	0.428
Waiting times	−0.003	0.115	−0.002	−0.024	0.981	−0.229	0.224
Infrastucture of facilities	0.257	0.163	0.111	1.575	0.116	−0.064	0.577
Cleanliness of the facilities	−0.232	0.207	−0.066	−1.122	0.262	−0.639	0.174
Proper signboards/direction guides	−0.004	0.108	−0.002	−0.037	0.971	−0.216	0.208

Notes: β Coef = Beta coefficient from the lineal general model, after adjustment by all relevant variables. Positive values indicate more satisfaction on that domain for that category; negative values indicate less satisfaction compared with the reference category, which is blank or indicated as “versus”. * Grading scale = 0–10.

## Data Availability

The data will be accessible from the corresponding author when the Ethics Committee of the National and Kapodistrian University of Athens and the 1st Regional Health Authority of Attica provide data access permission.

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
