# Peer review of "Valuing Outpatients’ Perspective on Primary Health Care Services in Greece: A Cross-Sectional Survey on Satisfaction and Personal-Centered Care"

_healthcare, 2024, doi:10.3390/healthcare12141427_

Round 1

Reviewer 1 Report

Comments and Suggestions for Authors

This is a very interesting study which presents original findings. The authors should be complimented for the novelty of their study, their proper methodology, and their large specimen. The quality of the statistical analysis is very good. The methods are adequately presented, and the conclusions are well-documented by the results. Comments for the authors:

1) Some grammatical and linguistic errors need to be fixed.

2) Was the questionnaire in Greek or another language? Were all the participants Greeks? If not, was this questionnaire validated for use in another language?

3) Are there any data about reasons for non-participation in the survey available? If so, they must be presented, to increase the strength and the validity of the results.

4) Why are the results, like the age of the participants, not presented in mean values?  In such a large sample, the range is not that practical. 

Comments on the Quality of English Language

Some minor linguistic and grammatical errors need to be fixed.

Author Response

Dear Reviewer, 

Comments and Suggestions for Authors

This is a very interesting study which presents original findings. The authors should be complimented for the novelty of their study, their proper methodology, and their large specimen. The quality of the statistical analysis is very good. The methods are adequately presented, and the conclusions are well-documented by the results.

Comments for the authors:

1) Some grammatical and linguistic errors need to be fixed.

Response: We followed the suggestion of reviewer, upgrades were made

2) Was the questionnaire in Greek or another language? Were all the participants Greeks? If not, was this questionnaire validated for use in another language?

Response: We followed the suggestion of reviewer, in page 3, in lines 113-124 we included the phrase “the questionnaire prepared in Greek by”. The participants had to be 18 years old and know the Greek language. As we know from Mr. Aletras, this questionnaire was not validated for use in another language. Moreover, we rewrite the paragraph in page 3, in lines 120-124 for be more appropriate.

As concerned our survey, individuals had to be 18 years of age and older, know the Greek language, be able to communicate and able to speak with other people (page 4, in lines 171-172). Also, 84.09% of the patient sample had Greek nationality and 15.91% other (most of them were Balkans) (page 5, in lines 205-206).

3) Are there any data about reasons for non-participation in the survey available? If so, they must be presented, to increase the strength and the validity of the results.

Response: Unfortunately, we are not able to provide stable information on the reasons for this non-participation among patients. The patient was free to withdraw at any time during the questionnaire completion process without being obliged to give reasons and we were obliged not to criticize him. Maybe the fear of violation of their anonymity or how they will be treated at the next appointment may have been inhibiting factors for not completing this questionnaire. From our side as researchers, we informed in writing on the questionnaire and additionally vocally the reasons for this survey and clearly displayed (on a badge) our name and the signs of the National and Kapodistrian University of Athens and the 1st Regional Health Authority of Attica. We also had the written approvals of the study protocol for anyone who wanted to see them. However, the response rate was 84.33% (1,012/1,200) which considered generally satisfying.  

 4) Why are the results, like the age of the participants, not presented in mean values? In such a large sample, the range is not that practical. 

Response: Generally, for all the results, there are mean values or at least referred in Tables. The reviewer must be referred in page 10, lines 274-283 where age groups refers in sub-scales. We followed the suggestion of reviewer, upgrades were made.

Comments on the Quality of English Language:

Some minor linguistic and grammatical errors need to be fixed.

Response: We followed the suggestion of reviewer, upgrades were made.

Additional information

Τhe reviewer noted that the methods could be further developed

Response: We followed the suggestion of reviewer. Upgrades were made in pages 3-4, lines 113-124, 140-142, 159-161

Thank you for understanding the difficulty of the whole undertaking of this manuscript and contributing positively to its upgrade.

Reviewer 2 Report

Comments and Suggestions for Authors

The article is devoted to the survey of a chosen set of outpatients Attika region of Greece. The purpose was to estimate the quality of primary care service in Greece.

The material is presented in a clear and easy-to-comprehend way. The full set of questions in the questionnaire is presented. The statistical treatment is satisfactory. Yet, I have several remarks on the paper. Following them may help make the article better.

Minor:

- Strength and Limitations should not constitute a separate chapter. It may be included in Discussion.

- In Discussion, the authors compare different EU countries. What does the UK have to do with the EU?

- In the article, COVID-19-related difficulties arise all of a sudden in Conclusions. That should not be. The authors may either introduce COVID-19 topic earlier, or discard it at all.

Major: The authors provide detailed data, but they themselves recognise that there are other works devoted to the topic. My question is: to which extent can the article be regarded novel? What does it add to improving primary health care service in Greece? To change the text properly, I recommend:

1) introducing the main concept right at the beginning of the paper, around which the authors are going to build their logic;

2) discussing their results primarily in the context of Greece. The comparison around the EU countries is not in the least new. The comparison may be strengthened by adding other dimensions to inscribe the Greece’s experience in the EU’s context or it may be omitted at all. Now it looks strange, as the core ideas of the text are still concerned with Greece.

To summarise, I recommend a Major Revision of the paper.

Comments on the Quality of English Language

A proofreading by a native speaker is recommended. Some phrases need revision

Author Response

Dear Reviewer,

Minor:

- Strength and Limitations should not constitute a separate chapter. It may be included in Discussion.

  Response: We followed the suggestion of reviewer.

- In Discussion, the authors compare different EU countries. What does the UK have to do with   the EU?

  Response: We followed the suggestion of reviewer.

- In the article, COVID-19-related difficulties arise all of a sudden in Conclusions. That should not be. The authors may either introduce COVID-19 topic earlier, or discard it at all.

  Response: We followed the suggestion of reviewer.

Major: The authors provide detailed data, but they themselves recognize that there are other works devoted to the topic. My question is: to which extent can the article be regarded novel? What does it add to improving primary health care service in Greece? To change the text properly, I recommend:

1) introducing the main concept right at the beginning of the paper, around which the authors are going to build their logic;

Response: The main concept of this manuscript refers to patient's evaluation in primary care centers in Athens Greece, after the introduction of a new structure model in primary health care, the national PHC network (PEDY) under the responsibility of the Health Authorities (DYPE) by geographical area, in 2014. The protocol of this survey with the questionnaire approved by Medical School of the National and Kapodistrian University of Athens in collaboration with the 1st Regional Health Authority of Attica (Ethical Permission). All approvals and the questionnaire have been sent to the journal.  There are strong phrases in introduction that provide our logic in this manuscript, more specifically:

  1. in lines 56-61: “Over the past 30 years, patients' satisfaction become an attribute of quality and has gained widespread recognition as a critical factor in strengthening healthcare system, since it provides as a useful insight into public opinion on healthcare system performance and responsiveness. Systematic reviews of published literature confirm that enhanced patient satisfaction not only results in improved patient experiences but also aligns with better treatment outcomes and lower health inequalities”
  2. in lines 82-91: “significant changes in primary care centers were initiated in the early 2010s, as many aspects needed to be improved [21-24]. In 2014, while the National Organization for the Provision of Health Services (EOPYY) had the responsibility of providing primary care and at the same time was the purchaser of these services, it was deemed appropriate to separate these responsibilities and to integrate the services of primary health care into the NHS and to set up the national PHC network (PEDY) under the responsibility of the Health Authorities (DYPE) by geographical area. A new PHC model needed to be developed based primarily on family medicine – a medical specialty with a wide range of competencies, similar to the solutions found in countries with strong PHC.”
  3. in lines 94-109: “The focus of the current study was to obtain in-depth information from patients about their perceptions of care and evaluate eight key factors that are commonly associated with the patient experience of PHC network (PEDYs), namely: arrival and admission, waiting before the appointment, cleanliness of toilets, medical examination and behavior of physician, behavior of nursing staff, laboratories, departure and the contribution of the PHC center, to provide information for: 1) how do Greek citizens perceive the quality of primary health care provided, 2) describe patient experiences with primary care, and 3) examine the association between primary care practices and patient satisfaction. Based on these findings, this study will help to fill the gaps in our knowledge by examining factors that may be associated with patient satisfaction in public primary health care facilities in Greece.”

In reality, this must be the first big survey after implimitation of the new PHC network (PEDY). In paragraph about the focus of the current study (page 3, in lines 99-110) “The focus of the current study was…European Union” we adjusted the wording accordingly hoping this make our purposes more clearly. Finally, even if had not introduced this structure model of primary care in Greece in 2014, systematic gathering of information on patients’ needs and experiences, using different methodologically-sound instruments, should be an integral part of routine care.

2) discussing their results primarily in the context of Greece. The comparison around the EU countries is not in the least new. The comparison may be strengthened by adding other dimensions to inscribe the Greece’s experience in the EU’s context or it may be omitted at all. Now it looks strange, as the core ideas of the text are still concerned with Greece.

Response: The focus of this research is the primary care in Greece indeed, as the survey, the questionnaire and all the participants are from Greece. However, we cannot ignore the fact that Greece belongs to the European Union. So, we should examine which systems are developed in other countries? What practices are they using to achieve better outcomes for primary care and their patients (e.g. high life expectancy)? How satisfied patients are with these systems and practices? What results have these practices produced in life expectancy of population?  

Bearing the above in mind, we consider that:

  1. the reader is aware of important data with just one photo and further reading the text acquires a general knowledge of patient satisfaction in Europe without having to refer to other articles.
  2. the comparison around the EU countries about patient satisfaction is not new, but it is constantly changing rapidly under differences in organizational, financial, regulatory, cultural and financial characteristics of a health system. Although a large number of studies have addressed patient satisfaction, a firm consensus regarding its determinants remains elusive. One reason is the lack of a common theoretical framework for patient satisfaction, and another is the fact that patient satisfaction is a complex and multidimensional concept with numerous determining factors.
  3. in order for Greece, through the adoption of good practices from other European countries, to reach high levels of operation of its systems, and for Greek patients to enjoy the same quality services and levels of satisfaction as the rest of European citizens.
  4. we believe that submitting articles to an international journal such as MDPI's Healthcare should refer to multi-country comparisons, texts that refer to a single country are more for local journals.

For this reason, we followed your recommendation to strengthened the comparison by adding and other dimensions.

Additional information:

Τhe reviewer noted that the introduction and conclusions must be improved, research design can be improved

Response: We followed the suggestions of reviewer. Major upgrades were made in introduction, page 3 in lines 96-99 and 107-110. Also, in conclusions, pages 7-8 in lines 517-525. 527-535 and 547-553. In research design, major upgrades were made in results and discussion section.

Thank you for understanding the difficulty of the whole undertaking of this manuscript and contributing positively to its upgrade.

Reviewer 3 Report

Comments and Suggestions for Authors

STRENGTHS

a) The foundational arguments of the paper are well researched and strong. 

b) The focus of the argument is well described.

c) The expected outcomes of the research are well described. 

SUGGESTIONS

a) Many paragraphs of the Results section are too content dense. It is difficult to follow the arguments made by the authors in the present form of the manuscript. SUGGESTION: Please consider breaking down these paragraphs into smaller/shorter paragraphs to facilitate ease of information absorption by the reader/scholar.

b) It is unclear why the authors needed to add in the section on the other health systems used in the EU. It is confusing as the Discussion does not directly address the issues that are raised in the Results section.

c) The authors then go on to write about Greece, but the Discussion on this is more political and far less about the data found in the Results section.

Therefore, the suggestion here is to revise the Discussion in it's entirety and to refocus on the Results and what do the Results tell us about the experience of Greek outpatients. In it's present form the Discussion section comes across more as a political diatribe than a Discussion of the data that was produced by the research. Thank you.

Comments on the Quality of English Language

Please proofread the article for very minor writing and grammatical errors.

Author Response

Dear Reviewer 3,

Comment: a) Many paragraphs of the Results section are too content dense. It is difficult to follow the arguments made by the authors in the present form of the manuscript. SUGGESTION: Please consider breaking down these paragraphs into smaller/shorter paragraphs to facilitate ease of information absorption by the reader/scholar.

Response: We followed the suggestion of reviewer, upgrades were made.

Comment: b) It is unclear why the authors needed to add in the section on the other health systems used in the EU. It is confusing as the Discussion does not directly address the issues that are raised in the Results section.

Response: The focus of this research is the primary care in Greece indeed, as the survey, the questionnaire and all the participants are from Greece. However, we cannot ignore the fact that Greece belongs to the European Union. So, we should examine which systems are developed in other countries? What practices are they using to achieve better outcomes for primary care and their patients (e.g. high life expectancy)? How satisfied patients are with these systems and practices? What results have these practices produced in life expectancy of population?

Bearing the above in mind, we consider that:

- the reader is aware of important data with just one photo and further reading the text acquires a general knowledge of patient satisfaction in Europe without having to refer to other articles.

- the comparison around the EU countries about patient satisfaction is not new, but it is constantly changing rapidly under differences in organizational, financial, regulatory, cultural and financial characteristics of a health system. Although a large number of studies have addressed patient satisfaction, a firm consensus regarding its determinants remains elusive. One reason is the lack of a common theoretical framework for patient satisfaction, and another is the fact that patient satisfaction is a complex and multidimensional concept with numerous determining factors.

- in order for Greece, through the adoption of good practices from other European countries, to reach high levels of operation of its systems, and for Greek patients to enjoy the same quality services and levels of satisfaction as the rest of European citizens.

- we believe that submitting articles to an international journal such as MDPI's Healthcare should refer to multi-country comparisons, texts that refer to a single country are more for local journals.

Οn the occasion of this observation of yours, the Discussion section was reorganized and we think that now is more clearly connected with the Results section: a) % of satisfied patients with primary care services, b) aspects of services considered as important (e.g. physician’s skills, medical equipment, waiting time, proximity, accessibility, communication, patient-centeredness, continuity, longer patient-physician relationship etc.), c) aspects of higher satisfaction with the services (e.g. empathic physicians, longer patient-physician relationship).

We followed the suggestion of reviewer, upgrades were made.

Comment: c) The authors then go on to write about Greece, but the Discussion on this is more political and far less about the data found in the Results section.

Therefore, the suggestion here is to revise the Discussion in it's entirety and to refocus on the Results and what do the Results tell us about the experience of Greek outpatients. In it's present form the Discussion section comes across more as a political diatribe than a Discussion of the data that was produced by the research. Thank you.

Response: We followed the suggestion of reviewer, upgrades were made. The Discussion section about Greece revised and refocus on the Results section.

Additional information

Τhe reviewer noted that the conclusions must be improved

Response: We followed the suggestion of reviewer. Upgrades were made in pages 7-8, lines 517-

536 and 547-548.

Thank you for understanding the difficulty of the whole undertaking of this manuscript and contributing positively to its upgrade.

Round 2

Reviewer 2 Report

Comments and Suggestions for Authors

Everything is good now. The paper can be published. No more comments.

Reviewer 3 Report

Comments and Suggestions for Authors

Thank you for the revisions and congratulations on your achievement.